# Thorax temperature and niche characteristics as predictors of abundance of Amazonian Odonata

**Lenize Batista Calvão[1,2], Ana Paula J. Faria[1,2], Carina Kaory Sasahara de Paiva[2], José Max Barbosa Oliveira-Junior[3], Javier Muzón[4], Alex Córdoba-Aguillar**  **[5]\*, Leandro Juen[1,2]**

**1** Programa de Pós-Graduação em Ecologia, Universidade Federal do Pará, Belém, Pará, Brazil, **2** Laboratório de Ecologia e Conservação (LABECO), Instituto de Ciências Biológicas (ICB), Universidade Federal do Pará (UFPA), Belém, Pará, Brazil, **3** Instituto de Ciências e Tecnologia das Águas (ICTA), Universidade Federal do Oeste do Pará (UFOPA), Santarém, Pará, Brazil, **4** Laboratorio de Biodiversidad y Genética Ambiental (BioGeA), Universidad Nacional de Avellaneda, Avellaneda, Buenos Aires, Argentina, **5** Departamento de Ecología Evolutiva, Instituto de Ecología, Universidad Nacional Autónoma de México, Ciudad Universitaria, Mexico City, Mexico

\* acordoba@iecologia.unam.mx

## Abstract

Environmental architecture and body temperature drive the distribution of ecto-thermic species, especially those with specific ecophysiological requirements or narrow ecological niches. In this study, we evaluated the connection between thorax temperature and niche specialization concerning the abundance and species contribution to the beta diversity of adult Odonata in Amazonian streams, employing the Species Contribution to Beta Diversity (SCBD). Our hypotheses were (i) Odonata species' thorax temperature is positively correlated with both morphology (thorax width) and air temperature and differ between suborders. (ii) The thorax temperature of the Odonata assemblage serves as a more influential predictor than niche specialization in determining species abundance and SCBD. We sampled 46 streams in an anthropized landscape in the Northeastern and Southeastern regions of Pará state, Brazil. Notably, niche breadth emerged as the variable influencing the abundance and SCBD of the Odonata assemblage. Niche position is nega-tively related with abundance only for Zygoptera. Anisoptera exhibited a negative relationship between abundance and thoracic temperature. On the other hand, Zygoptera had positive relationship between abundance and thoracic temperature. In summary, our results underscore the necessity of considering both niche and ecophysiological predictors to comprehensively assess the Odonata assemblage in Amazonian streams. This holistic approach has implications for conservation efforts and bioassessment practices, offering valuable insights into the collective response of Odonata as a group.

**Data availability statement:** Data can be accessed for free from: https://doi.org/10.6084/m9.figshare.28686089.

**Funding:** We are grateful to the Hydro Paragominas Company for supporting the research project "Monitoring Aquatic Biota of Streams on Areas of Paragominas Mining SA, Pará, Brazil" (Process 011); the Biodiversity Research Consortium Brazil-Norway (BRC), for providing funding and logistical support, as well as for scholarship grants to authors LBC, and CKSP. We would also like to thank Amplo Engenharia e Gestão de Projeto for logistic support, and we are grateful to Vale Company for allowing our data collection within its operation area in the Carajás region. We thank Conselho Nacional de Desenvolvimento Científico e Tecnológico (CNPq) for scholarship grant for LBC (Process 154761/2018-4 and 168384/2022-1), and a research productivity grant to LJ (Process 304710/2019-9). APJF is grateful to the Conselho Nacional de Desenvolvimento Científico e Tecnológico (CNPq) for granting a postdoctoral scholarship (Process No. 306103) and Fundação de Amparo à Pesquisa do Estado do Piauí (FAPEPI) for providing funding a research Project.

**Competing interests:** The authors have declared that no competing interests exist.

## Introduction

One aim in ecology is to understand how species assemblages distribute as a function of each species' requirements. In this regard, ecological niche breadth and position are key predictors for the local abundance [1–4]. The specific link that underlies species abundance and such predictors is explained according to two models. The niche breadth hypothesis suggests that generalist species, with greater tolerance to environmental variations, tend to have higher abundance, whereas the niche position hypothesis argues that species in broadly available habitats are also more abundant [5].

Generalist species, endowed with greater niche breadth, can occur in a wider range of environmental conditions, leading to a smaller contribution of species to Beta diversity (SCBD) than with the contribution of specialist species [6]. Accordingly, niche position can influence SCBD, as species in marginal habitats use more specific environmental conditions than non-marginal species [6]. Simultaneously, the relationship of functional traits (e.g., thermoregulatory ability governed by body size) with SCBD may occur if traits confer adaptations in the species, influencing their abundance and/or occupancy at the site [2,4].

Odonata are aquatic insects whose thermal performance determines their distribution both at the micro- [7] and macro-scale [8]. In this context, the body size and behavior of adult odonates are associated with their thermoregulation strategies, categorizing species as thermal conformers, heliothermic or endothermic [7]. These thermoregulatory strategies are correlated with body size, as heat exchange with the environment occurs based on the surface/volume ratio [9]. Consequently, small percher species (e.g., most zygopterans) generally exhibit thermal conformers or heliothermic, relying on air temperature to start their activities [7]. These species can regulate heat loss by adjusting their body posture in response to light [10–11] and by selecting habitats that support their thermal requirements [9,12]. Conversely, larger species of Odonata can inhabit open areas with reduced canopy cover [7] as they are heliothermic [13], benefiting from direct sunlight exposure. Finally, larger species (e.g., mostly Anisoptera) are predominantly endothermic, enabling them to overcome the limitations faced by thermal conformers, as they can internally control heat loss or production through their thorax muscles [9].

Ecophysiological requirements have frequently been crucial predictors for odonate abundance or distribution in tropical streams [13–16]. This correlation is primarily supported by the impact of changes in vegetation cover and air temperature, both acting as environmental filters influencing odonate community composition [16]. For example, the absence of vegetation and reduced shade favor most anisopteran species due to their reliance on increased light input into the stream [7,13,15–17]. Conversely, these environmental conditions are unsuitable for most Zygoptera or smaller species, as they typically prefer habitats with a more consistent temperature and some species could overheat in altered environments with high air temperature and solar incidence [15–17].

In the present study, we aimed to evaluate the importance of thoracic temperature and niche specialization on the abundance and contribution of each species to the SCBD of adult Odonata in Amazonian streams. For this, we had the following

predictions: (i) thoracic temperature of Odonata species differs between suborders and correlates positively with morphology (thorax width) and air temperature. Larger species (Anisoptera) may exhibit heliothermic behaviors, leading to thoracic temperature higher than air temperature. Conversely, smaller species (Zygoptera) tend to thermoregulate in response to air temperature, resulting in thorax temperature like the ambient environment; (ii) the thoracic temperature of suborders (Zygoptera and Anisoptera), serves as an important predictor for abundance and SCBD given that eco-physiological traits play a crucial role in Odonata habitat selection. We expect a positive relationship between thorax temperature, niche breadth and abundance for Anisoptera and Zygoptera. Zygoptera conformers, with lower thoracic temperatures, are specialist species in environmental conditions to not overheat, and those species with higher temperatures close to or above ambient tend to be more abundant and have higher niche breadth. For Anisoptera, large species are more heliothermic/endothermic and can actively choose their microhabitat and be more abundant in environments suited to their activities and ecophysiological requirements. Niche position and abundance, for both suborders, would be a negative relationship because non-marginal species that occur in more habitat's conditions would have higher abundance.

For SCBD we expected a positive relationship between thorax temperature for Anisoptera, because endothermic species can adjust their body temperature and select the most favorable environment. On the other hand, Zygoptera will have a negative relationship with SCBD, because conformer species depend on specific habitat conditions and tend to be very sensitive to environmental changes and hence are more habitat specialists. Also, we expected a negative relationship for SCBD and niche breadth, and positive relationship between niche position and SCBD, for both suborders Fig 1.

## Materials and methods

### Study area

The study was conducted in 46 streams across four municipalities in the Northeastern region (Tomé-açu, Ipixuna do Pará, Concórdia do Pará and Acará) and three municipalities in the Southeastern region (Paragominas, Canaã dos Carajás and Parauapebas) of Pará state, Brazil, covering basins ranging from the Tocantins-Araguaia River and Capim River (Fig 2, Supplementary Material Table S1). Northeastern Pará is characterized, according to the Köppen classification, by a tropical rainforest climate (Af) and tropical monsoon climate (Am) [18], with temperature ranging from 22°C to 34°C (minimum: 22°C to 23°C; maximum: 30°C to 34°C). Southeastern Pará has a Savanna climate, classified as a tropical climate with a dry season (Aw) [18–20]. The municipalities of Canaã dos Carajás and Parauapebas, located in Serra dos Carajás area, are notable for their landscape, mainly due to their elevation, which ranges between 500 and 700 m a.s.l. [19]. The average monthly temperature in this area varies from approximately 25°C to 26°C, with an annual rainfall of 2.033 millimeters [19].

### Insect sampling

The 46 sampled streams ranged from 1st to 3rd order, with an average width of 2.8 meters and a depth of 29.9 centimeters, according to Strahler [21] classification. Sampling periods were performed between July 2017 and October 2018, consistently during the low precipitation period. Adult odonates were collected only once stream, in 20 segments spaced 5m apart, distributed along a continuous 150-meter longitudinal stretch in each stream (further details in [15,22,23]). Specimens were collected on stream banks for one hour, always between 11:00 and 14:00 hrs (the peak activity period for adult odonates), at temperatures above 19°C, using a 40 cm diameter and 65 cm in length entomological net [16,24]. Standardization of sampling effort and climatic conditions was necessary due to the organisms on solar incidence [25], ensuring the presence of all ecophysiological groups [16].

The collected specimens were placed in tracing paper envelopes and subsequently fixed in acetone P.A. (Propanone) for 12 hours for Zygoptera and 48–72 hours for Anisoptera. Species identification was conducted to the species level using specialized taxonomic keys [26–30]. The biological material was deposited in the Collection of Aquatic Insects at the Laboratory of Ecology and Conservation, Federal University of Pará (UFPA), University Campus of Belém, Pará, Brazil [31].

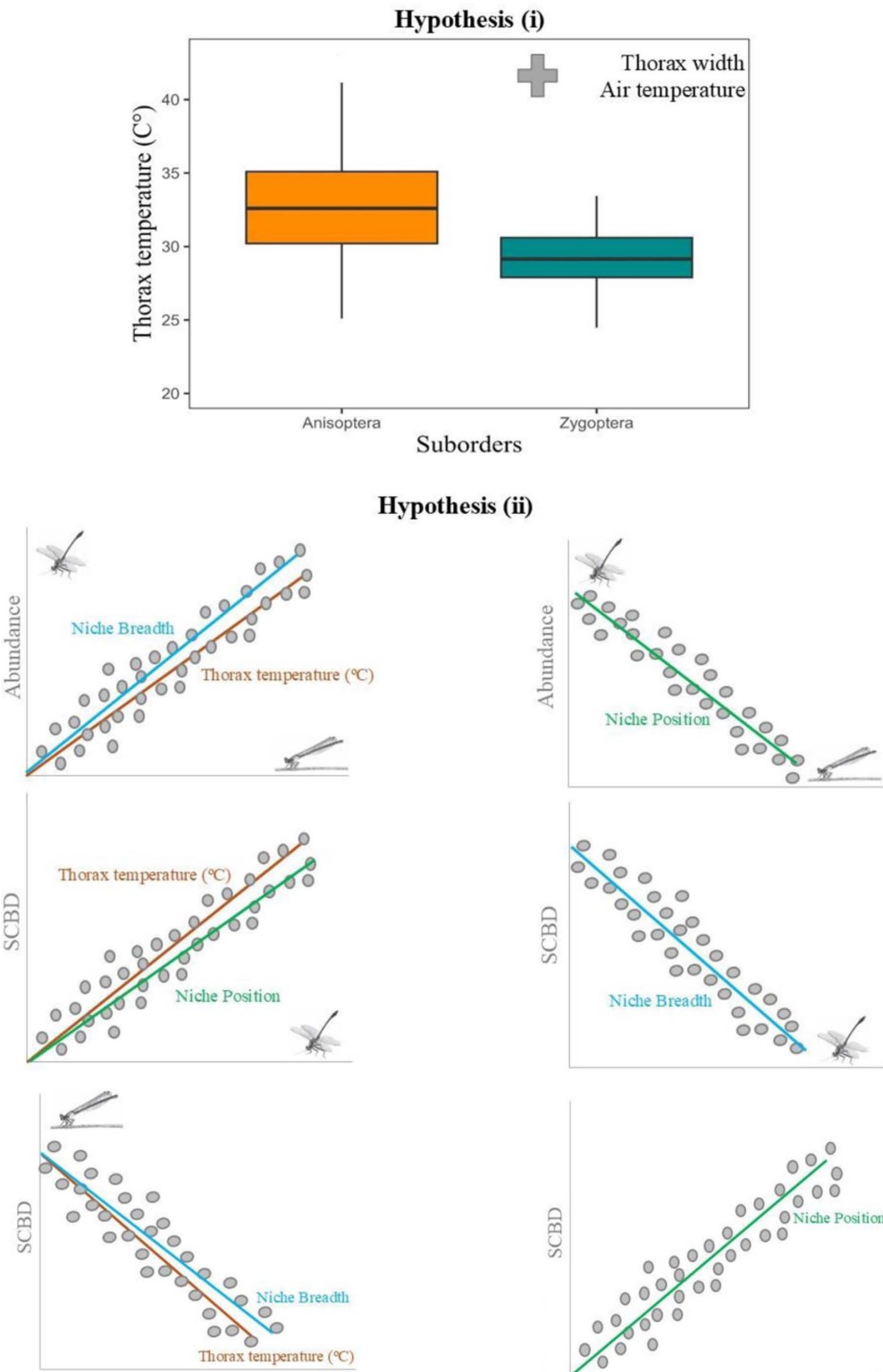

**Fig 1.  Schematic drawing of hypotheses and images of Odonata were adapted from Stehr, F.W and Tennessen, K.J.**

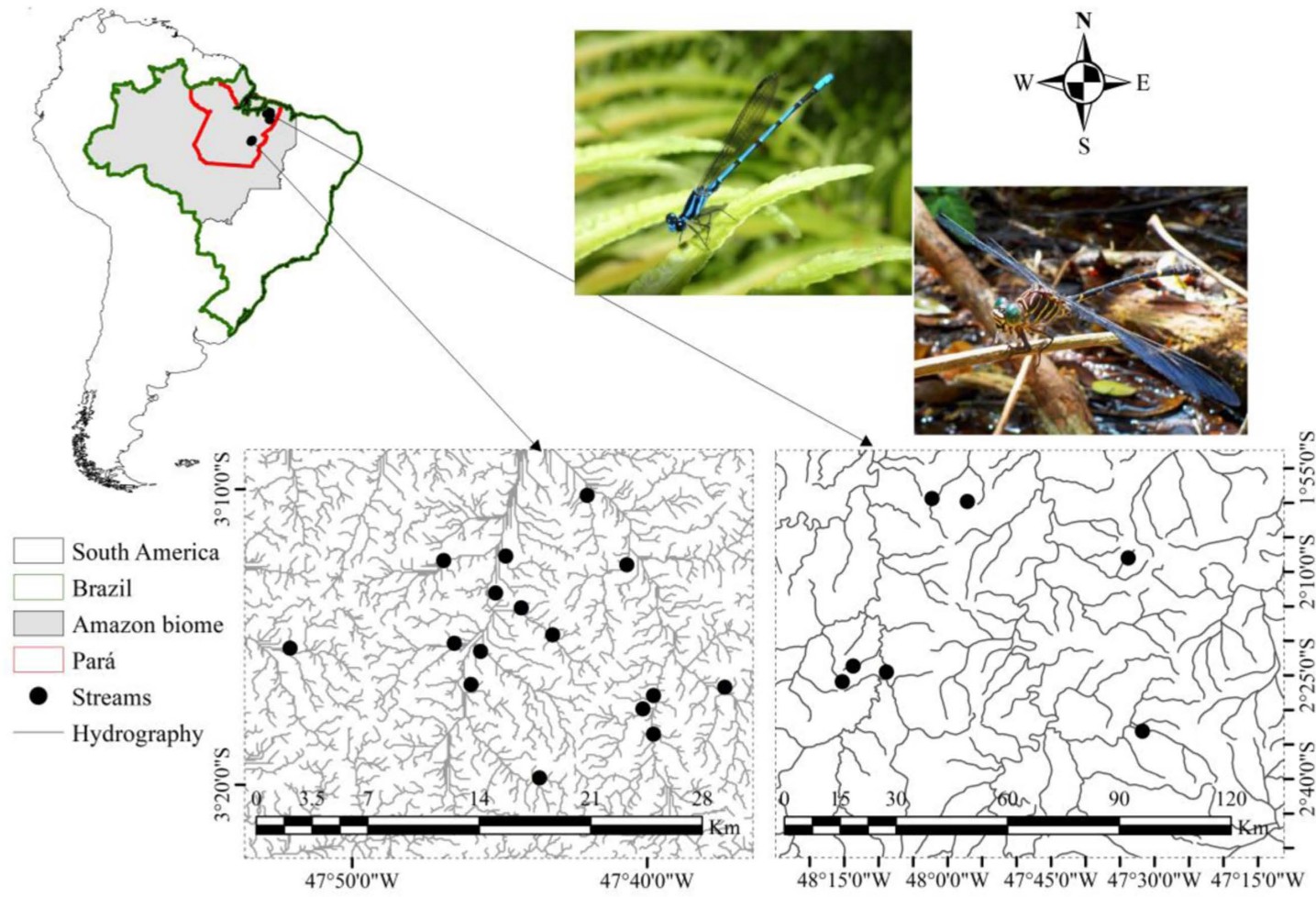

**Fig 2. Study area showing the 46 streams distributed in the Northeastern and Southeastern of Pará (Red), Brazil (Green). Images of Odonata were taken by L. Juen. Map was generated using QGIS.**

## Measurements of physiological traits

The thoracic temperature of males was captured using an entomological net and quickly measured within a maximum of 10 seconds to avoid physical damage and alteration in body temperature. We used only male individuals because species identification is possible in this sex. Holding the wings, we measured the body temperature (°C) of each individual using a Testo-805 infrared thermometer (accuracy ± 1°C [−2 to +40°C], resolution 0.1°C [−9.9 to +199.9°C], and reaction time < 1s). The recording was performed by pointing the thermometer beam at the center of the right side of the thorax, at a distance of five centimeters.

Thorax width was measured using a digital caliper (accuracy = 0.02 mm) for males of each species. For species with multiple individuals, we used the averages, while for those with only one individual, we recorded the absolute value.

## Environmental variables in streams

For each stream, four environmental variables were measured: depth (cm), channel width (m), Habitat Integrity Index (HII) and air temperature (in ºC). Depth was measured at three points of the stream: left and right of the bank, and

central region. The average of these measurements was used as a measure of depth in each sampling unit. The HII was composed by 12 items that describe the degree of habitat integrity in the stream: land use pattern beyond the riparian zone; width of riparian forest; completeness of riparian forest, vegetation of riparian zone within 10 m of channel; retention devices and sediment in the channel; river bank structure; bank undercutting, stream bottom; distribution of riffles and pools; characteristics of aquatic vegetation and detritus [32]. HII ranges from 0 (altered stream) to 1 (preserved stream), according to the habitat integrity conditions found in each stream [32]. Note that this HII has been widely used in studies that have assessed the environmental conditions of streams and their relationship with aquatic insect diversity [33–35]. Air temperature was measured using a Testo-805 infrared thermometer. After measuring the thorax temperature of the specimens, the thermometer was positioned in the microhabitat where each individual was collected.

## Data analysis

To calculate niche breadth and position (i.e., predictor variables) of the species, we used the Outlying Mean Index (OMI) [5] which is based on the following environmental variables: depth, channel width, and HII. The OMI analysis calculates the distance between the average environmental conditions used by the species (centroid) and the average environmental conditions of the sampled sites (hyperspace) [5,36]. Species abundance data were logarithmized and environmental variables standardized. The marginality significance (OMI) of the species was evaluated using the Monte-Carlo permutation test with 1000 permutations [5]. Thus, we obtained the breadth (environmental tolerance) and position (marginality) of the ecological niche for each species. We carried out PCoA for Odonata composition visualization using Hellinger and Bray-Curtis.

The ecological uniqueness of species (SCBD) was measured from the total variation of the assemblage (Total Beta Diversity – $BD_{Total}$) between streams. For this analysis, we submitted the Odonata composition matrix to the Hellinger transformation, which was used to measure the Sum of Total Squares ($SS_{Total}$). Then, we divided the $SS_{Total}$ by the number of sampled streams (n-1) and obtained the $BD_{Total}$ of the assemblage that was partitioned into ecological uniqueness of species (SCBD) [37]. For more details on the calculation of $BD_{Total}$ and SCBD, see [37]. Finally, species with higher SCBD values had a higher relative contribution to beta diversity [37].

We performed the analysis of the relationship between the variables from the generalized linear mixed models (GLMMs) [38] and species as random factors. Thus, to assess (i) the difference in thoracic temperature of Anisoptera and Zygoptera, the relationship between the interaction of air temperature and thorax width on Odonata thoracic temperature, and the relationship between the difference of air and thoracic temperature (Tth-Tair [ºC]) and thorax width for Anisoptera and Zygoptera, with Gaussian distribution. The analyzed models contained each individual as a sample unit.

To assess the effect of thoracic temperature, niche breadth and position, predictor variables were standardized using decostand function (method = 'standardize') from the vegan package in R. For Suborders, we carried out a GLM (Negative Binomial). The analyzed models contained species as a sample unit. We evaluated overdispersion and performed the visual validation of the models using the simulated envelope [39]. To assess the effect of thoracic temperature, niche breadth and position (predictor variables were standardized using decostand function). Also, we assessed collinearity, and we did not detect high correlation [> 40] values between the predictor variables before running the models. Only niche breadth and abundance are positively related in 79. For Odonata suborders SCBD (response variable), we used a Beta Regression [41]. This analysis is more suitable for the response variable (SCBD) distributed between values from 0 to 1 [41]. The binding function used in this model was logit.

All analyses were performed using the R software [42] using MASS [43], hnp [39], betareg [41], and vegan packages [44].

## Results

### Odonata assemblage

856 specimens were collected, belonging to 54 species: 22 anisopterans and 32 zygopterans (Supplementary Material Table S2). The data were distributed in six Zygoptera families and one Anisoptera family. The most abundant species of Zygoptera were *C. rutilans, M. aenea* and *E. metallica*. For Anisoptera were *F. amazonica, E. basalis* and *P. lais*.

### Thoracic temperature and size, and their relation with ambient temperature

Anisopterans showed greater temperature and thorax width than zygopterans. On average, Anisoptera species have 3°C more than Zygoptera. Air temperature was higher at least 1°C in the places where Anisoptera species were sampled (Tables 1–2 and Fig 3). Both thoracic width and air temperature affect thoracic temperature of Odonata individuals (Table 2).

Anisopterans had an average difference of thoracic and air temperature of approximately 2°C above the air temperature. For Zygoptera, the same pattern was observed with an average difference of approximately 1°C below air temperature (Table 3 and Fig 4). Both relationships have a positive relationship with thorax width.

### Stream structure and odonate assemblage

There was a relation between environmental variables (depth, channel width and HII) and Odonata species (p = 0.010) (OMI analysis global test). The variable that contributed the most to the first sorting axis was HII (0.88), followed by width (−0.22) and depth (0.03). This variable is essential to assess environmental integrity and demonstrates that Odonata species change composition with more intact streams (HII above 0.7) and with multiple anthropic activities (Fig 5). Streams with greater habitat integrity have a greater number of species of Anisoptera and Zygoptera that are only collected in these environments (Supplementary MaterialTable S3).

**Table 1. Mean and standard deviation (SD) values of thorax temperature, air temperature, and thorax width for Zygoptera and Anisoptera sampled in Brazilian Amazon streams.**

**Anisoptera**

|  | Thorax width (cm) | Thorax temperature (ºC) | Air temperature (ºC) |
|---|---|---|---|
| Mean | 2.669 | 32.657 | 30.986 |
| SD | 0.960 | 3.568 | 2.620 |

**Zygoptera**

|  | Thorax width (cm) | Thorax temperature (ºC) | Air temperature (ºC) |
|---|---|---|---|
| Mean | 1.507 | 29.172 | 29.906 |
| SD | 0.470 | 2.230 | 1.891 |

**Table 2. Results of the GLMM (Gaussian distribution) (species random effect) evaluating the relationship between thorax temperature of Odonata (response variable) and suborders and interaction of air temperature and thorax width.**

|  | Value | Std.Error | DF | t-value | p-value |
|---|---|---|---|---|---|
| Intercept | 29.780 | 0.260 | 523 | 114.464 | **<0.001** |
| Suborders | 1.694 | 0.506 | 52 | 3.345 | **0.002** |
| Thorax width (cm) | 0.844 | 0.170 | 523 | 4.956 | **<0.001** |
| Air temperature (ºC) | 1.142 | 0.096 | 523 | 11.867 | **<0.001** |
| Thorax width *Air temperature | 0.302 | 0.086 | 523 | 3.507 | **0.001** |

Significant p values appear in bold.

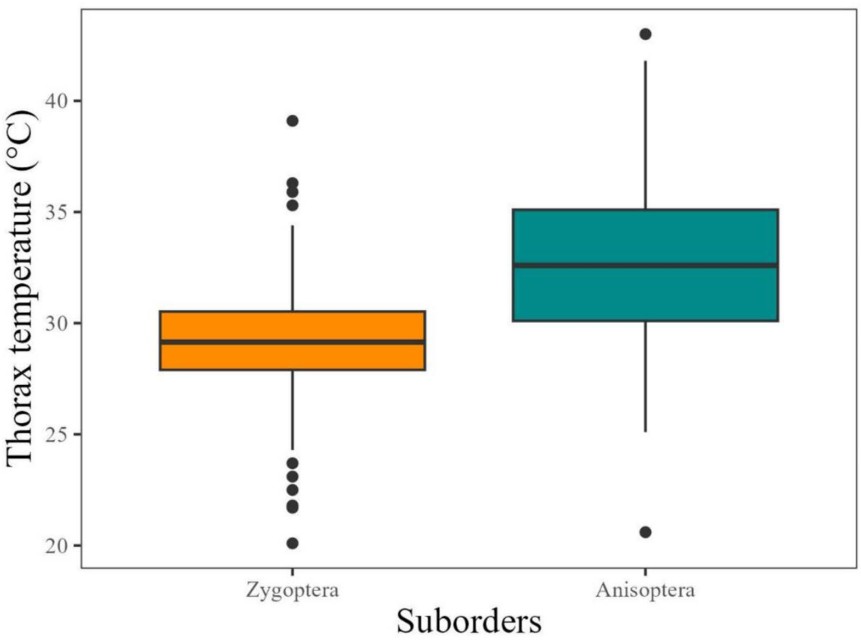

**Fig 3. Thoracic temperature of Anisoptera (Dark orange) and Zygoptera (Cyan). Black dots are outliers.**

**Table 3. GLMM (Gaussian distribution) (species random effect) results evaluating the difference between air and thorax temperature (Tth-Tair – variable response) with thorax width of Anisoptera and Zygoptera (predictor).**

**Zygoptera**

|  | Value | Std.Error | DF | t-value | p-value |
|---|---|---|---|---|---|
| Intercept | −2.230 | 0.496 | 331 | −4.493 | **<0.001** |
| Thorax width (cm) | 0.972 | 0.299 | 331 | 3.250 | **0.001** |

**Anisoptera**

|  | Value | Std.Error | DF | t-value | p-value |
|---|---|---|---|---|---|
| Intercept | −0.908 | 1.045 | 193 | −0.869 | 0.386 |
| Thorax width (cm) | 0.929 | 0.290 | 193 | 3.203 | **0.002** |

Significant p value appears in bold.

## Abundance of Odonata, SCBD, and niche and thorax temperature as predictors

The most abundant species of Zygoptera were *C. rutilans*, *M. aenea* and *E. metallica*. For Anisoptera were *F. amazonica*, *E. basalis* and *P. lais*. The average the niche breadth for Zygoptera was 0.274 (standard deviation ± 0.334) and the position 1.540 (±1.164). For Anisoptera 0.165 (±0.214) and 3.012 (±2.265). Species with greater niche breadth were *M. cupraea*, *H. silvarum* and *M. aenea* (Zygoptera) and *E. fusca*, *O. walkeri* and *O abbreviata* (Anisoptera). For niche positions were *A. luteum* and *H. icterops*, *Argia fumigata* (Zygoptera) and *D. obscura*, *B. herbida* and *E. cannacrioides* (Anisoptera).

Niche breadth and thorax temperature were important predictors of suborders abundance (Table 4 and Fig 6). Niche position emerged as predictor only for Zygoptera.

On average, SCBD was 0.017 (standard deviation ± 0.026). Species with greater SCBD were *C. rutilans*, *H. indeprensa* and *M. aenea* for the suborder Zygoptera and *E. basalis*, *O. abbreviata* and *F. amazonica* for the suborder Anisoptera. When evaluating predictor variables for SCBD, only niche breadth emerged as predictor (Table 5 and Fig 6).

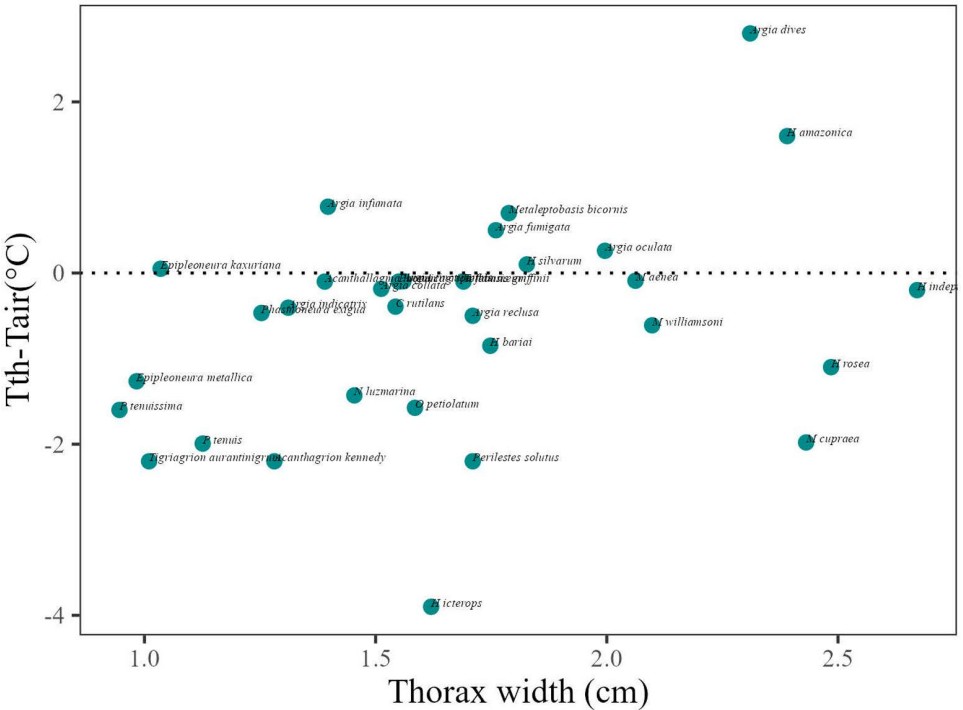

**Fig 4. Relation of the difference between air and thorax temperature (Tair-Tth (ºC) response variable) and thorax width (cm), for Anisoptera (Dark orange) dots bellow dotted line are species that have the thorax temperature above air temperature. Zygoptera (Cyan). Images of Odonata from Stehr, F.W and Tennessen, K.J.**

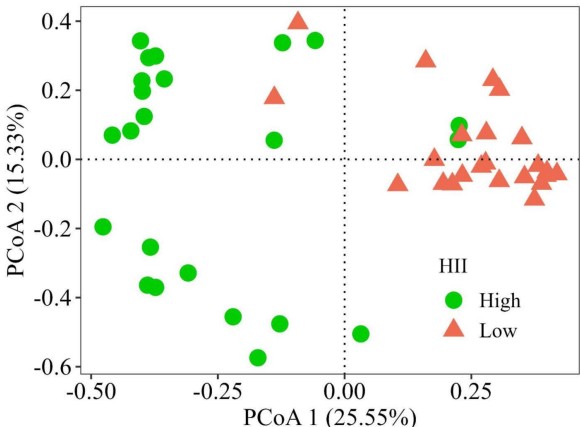

**Fig 5. Ordination (PCoA) of Odonata species and habitat integrity index (HII): green dots are HII above 0.7 and Coral dots are streams with multiple anthropic activities (HII bellow 0.7).**

## Discussion

Niche breadth was an important predictor for abundance and SCBD for Anisoptera and Zygoptera, as we expected. Our hypothesis was partially corroborated, niche position is negatively related with Zygoptera abundance and not for

**Table 4. GLM for suborders evaluating the relationship of species abundance with niche breadth, position and thorax temperature.**

**Anisoptera**

| | Estimate | Std. Error | z value | Pr(>\|z\|) |
|---|---|---|---|---|
| Intercept | 2.485 | 0.261 | 9.517 | **<0.001** |
| Thorax temperature (ºC) | −0.643 | 0.226 | −2.840 | **0.005** |
| Niche position (NP) | −0.236 | 0.215 | −1.094 | 0.274 |
| Niche breadth (NB) | 0.868 | 0.305 | 2.849 | **0.004** |

**Zygoptera**

| | Estimate | Std. Error | z value | Pr(>\|z\|) |
|---|---|---|---|---|
| Intercept | 2.698 | 0.315 | 8.578 | **<0.001** |
| Thorax temperature (ºC) | 1.027 | 0.418 | 2.458 | **0.014** |
| Niche position (NP) | −0.881 | 0.357 | −2.468 | **0.014** |
| Niche breadth (NB) | 0.846 | 0.179 | 4.714 | **<0.001** |

Significant p values appear in bold.

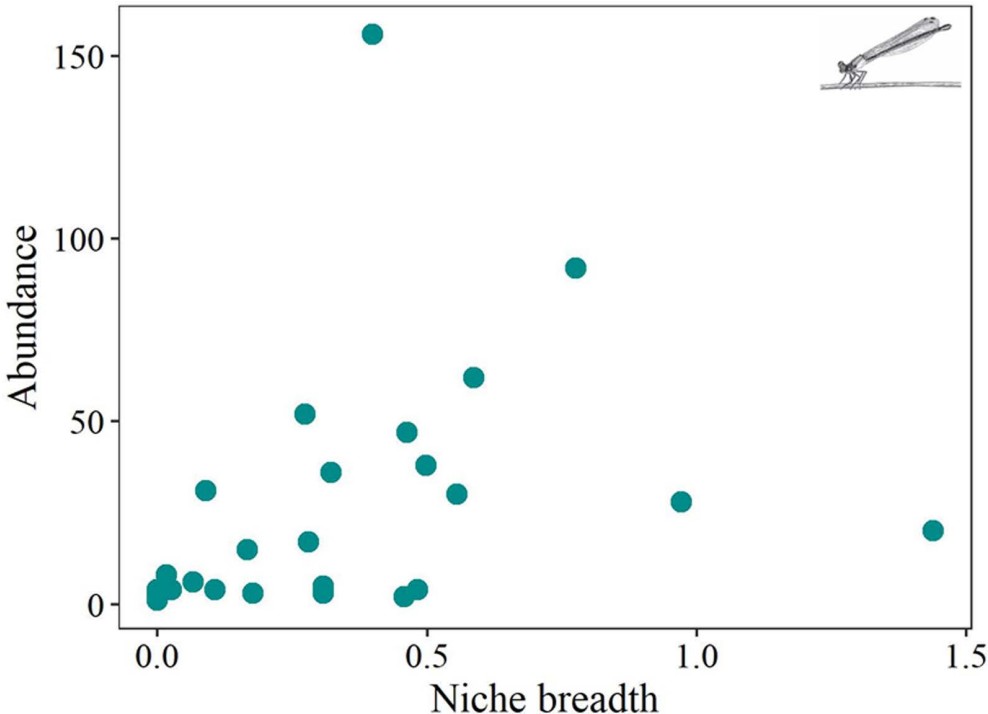

**Fig 6. Significant relationships of Abundance and SCBD of the Odonata suborders and niche breadth, position and thorax temperature.** Anisoptera (Dark orange) and Zygoptera (Cyan). Images of Odonata from Stehr, F.W.

Anisoptera. Also, the thoracic temperature was a good predictor only for abundance and not for SCBD. Research shows that species with greater niche breadth are more abundant and tend to present greater regional occupation [45]. Furthermore, these species often display a more generalist or tolerant behavior towards diverse environmental conditions, which may contribute to their geographical range size [46] and maybe to the reduced vulnerability. In Odonata, this macroecological pattern may be due to the generalist species being able to persist in degraded environments [47], resulting

**Table 5. Beta regression results between SCBD of suborders and niche breadth, position, and thorax temperature.**

**Anisoptera Pseudo R-squared: 0.485**

|  | Estimate | Std. Error | z value | Pr(>|z|) |
|---|---|---|---|---|
| Intercept | −4.357 | 0.237 | −18.363 | **<0.001** |
| Thorax temperature (ºC) | −0.152 | 0.180 | −0.846 | 0.397 |
| Niche position (NP) | −0.190 | 0.163 | −1.170 | 0.242 |
| Niche breadth (NB) | 0.563 | 0.232 | 2.423 | **0.015** |
| **Zygoptera Pseudo R-squared: 0.492** | | | | |
| Intercept | −4.000 | 0.286 | −13.996 | **<0.001** |
| Thorax temperature (ºC) | 0.136 | 0.332 | 0.410 | 0.682 |
| Niche position (NP) | −0.444 | 0.263 | −1.689 | 0.091 |
| Niche breadth (NB) | 0.376 | 0.113 | 3.323 | **0.001** |

Significant p values appear in bold.

in a broader distribution range [48]. In our study the families Libellulidae and Coenagrionidae, are more abundant and both are diverse families at a continental scale which, which may explain their wide distribution [49,50]. Some species of Coenagrionidae are associated with degraded environments in previous study [40]. More specifically within this family *E. metallica* and *T. aurantinigrum,* and libelluideos *E. fusca, E. basalis* and *O. walkeri* are associated with streams with environmental change [51–52]. In addition, other species such as *M. cupraea, H. silvarum* and *M. aenea* presented greater niche breadth. Although species with lower abundance and smaller niche breadth tend to be sensitive to environmental changes, which make them less likely to persist when the physical characteristics of streams are modified by anthropogenic actions [24], we highlight in Zygoptera, that even species such as *M. aenea and C. rutilans* in Amazonian streams seem to have limits to occur in different intensities of multiple impacts. Consequently, these species can disappear with impacts that lead to drastic changes in the streams [52].

Only for Zygoptera we have found niche position negatively associated with abundance indicating that species, when very abundant, are non-marginal and occur in more common habitats in the region, including altered streams. Marginal species with high niche positions such as *A. luteum* are specialized in Amazon rainforest streams and use leaves of perch or habitat above-water habitats to remain standing [29]. Even though in our study, it occurred in a stream with less environmental integrity, it is possible that some parts of vegetation important for *A. luteum*, remain under anthropogenic impact and are still able to maintain some characteristics important for it. Another species with the same pattern were *H. icteropts,* even though they are found in environments with low integrity, it has high specialization, and these environments still have habitats that favor their presence. In our study, this species seems to be very dependent on air temperature and maintains temperatures up to around 4 °C below air temperature, indicating that the presence of riparian vegetation is very important to maintain an adequate microclimate. *A. fumigata* had high niche position and depends on areas with riparian vegetation and occurred in our study in streams with greater habitat integrity.

Contrary to our hypothesis, niche characteristics appear more important for contribution of these species to beta diversity than body temperature. In fact, much previous work has demonstrated how environmental filters affect Odonata diversity and their microhabitat structure (Bank angle, wood in the stream bed), physical and chemical, canopy cover [51,53] and regional variables surroundings streams [17]. Species with greater niche breadth had a greater contribution to beta diversity, which is the opposite to expected [45,46]. Possibly more habitat generalist species and adults that are active dispersers and can survive in a wide range of environmental conditions play an important role in governing SCBD. Previous studies show that the pattern of beta diversity of an assemblage is also adequately described by the common species, according to [54].

Ecophysiological traits as body size and thorax temperature, therefore, provides additional insights into the patterns of suborders abundance and contribute to metacommunity dynamics due to its thermoregulatory restrictions [13]. There is a negative relationship between thoracic temperature and abundance of Anisoptera, contrary our expectative. In general, Anisoptera can heat their bodies through heliothermic or even endothermic ability. These thermoregulatory abilities are crucial for Odonata distribution [13]. Furthermore, heliothermic anisopterans can benefit from habitats with reduced riparian vegetation and greater sunlight input and can be quite abundant in these areas [16]. Conversely, larger species of Anisoptera (e.g., Gomphidae) with higher body temperatures may be endothermic. These species live inside forests and are often found in streams only when they arrive for mating and oviposition [55]. These ecological and behavioral traits make this species, which has higher thoracic temperatures, more difficult to collect. Zygoptera with higher thoracic temperatures tend to be more abundant than thermal conformers. However, more abundant species are those with greater niche breadth which are more tolerant to environmental conditions.

Thoracic temperature differs between suborders and is related to thorax width and air temperature. Anisopterans presented an average thoracic temperature up to 2ºC above air temperature, showing a negative relationship with thorax width. For Zygoptera maintains the thorax temperature 1ºC below air temperature. Differences in the behavior of these groups may help explain this pattern. Anisoptera tend to be heliothermic or endothermic fliers and maintain the thoracic temperature above that of the air. One of the ways to control heat loss is to alter the circulation of hemolymph between the thorax and abdomen [10] posture adjustment [9]. Zygoptera can maintain the temperature of the thorax closer to the air, previous studies demonstrate that this difference can be up to 1ºC [56]. The smaller ones tend to be thermal conformers, and the larger ones can be heliothermic, or even a continuum that may exist between these groups [57] and future studies can better investigate these categories.

Finally, niche characteristics may be important for the distribution of Odonata. Ecophysiological traits also were important for Anisoptera and Zygoptera abundance. May [10] suggests also that climate, body size and behavior are essential for maintaining the body temperature of Odonata. Changes in streams due of anthropic activities alter microclimatic patterns such as air temperature, fundamental for the physiological processes of Odonata species, leading to a change in the composition of species in these environments [33]. We have demonstrated also that adult odonate species composition varies in relation to habitat integrity. We therefore suggest that their monitoring would provide a good indicator of riparian zone quality considering niche characteristics and their thermoregulation abilities.

## Supporting information

**Table S1. Environmental information of the 46 streams sampled in the Northeastern and Southeastern of Pará, Brazil.**
(DOCX)

**Table S2. Odonata species. SD stands for standard deviation.**
(DOCX)

**Table S3. Odonata that are only collected in streams with higher (>0.7) and low (0.7) habitat integrity index (HII).**
(DOCX)

## Acknowledgments

We would like to thank Amplo Engenharia e Gestão de Projeto for logistic support, and we are grateful to Vale Company for allowing our data collection within its operation area in the Carajás region. Thanks Ana Luisa Fares, Naiara Raiol, Ana Luiza Andrade, Erlane José Cunha, Fernando Geraldo de Carvalho, for helping us with the biological sampling.

## Author contributions

**Conceptualization:** Lenize Batista Calvão, José Max Barbosa Oliveira-Junior, Javier Muzón, Alex Córdoba-Aguilar, Leandro Juen.

**Data curation:** Lenize Batista Calvão, Leandro Juen.

**Formal analysis:** Lenize Batista Calvão, Ana Paula J. Faria, Leandro Juen.

**Funding acquisition:** Leandro Juen.

**Investigation:** Lenize Batista Calvão, Ana Paula J. Faria, Carina Kaory Sasahara de Paiva, José Max Barbosa Oliveira-Junior, Leandro Juen.

**Methodology:** Lenize Batista Calvão, Ana Paula J. Faria, Carina Kaory Sasahara de Paiva, José Max Barbosa Oliveira-Junior, Javier Muzón, Alex Córdoba-Aguilar, Leandro Juen.

**Project administration:** Leandro Juen.

**Resources:** Leandro Juen.

**Supervision:** Leandro Juen.

**Validation:** Lenize Batista Calvão.

**Writing – original draft:** Lenize Batista Calvão, Ana Paula J. Faria, Carina Kaory Sasahara de Paiva, José Max Barbosa Oliveira-Junior, Javier Muzón, Alex Córdoba-Aguilar, Leandro Juen.

**Writing – review & editing:** Lenize Batista Calvão, José Max Barbosa Oliveira-Junior, Alex Córdoba-Aguilar, Leandro Juen.

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
