## [Decision Letter · Decision Letter 0]

PONE-D-24-40484Thorax temperature and niche characteristics as predictors of abundance of Amazonian OdonataPLOS ONE

Dear Dr. Córdoba-Aguilar,

Thank you for submitting your manuscript to PLOS ONE. After careful consideration, we feel that it has merit but does not fully meet PLOS ONE’s publication criteria as it currently stands. Therefore, we invite you to submit a revised version of the manuscript that addresses the points raised during the review process.

The reviewers have provided constructive feedback on your study. While your manuscript offers valuable insights into the ecological and physiological traits of Amazonian Odonata, several critical areas require revision to ensure clarity, rigor, and alignment with the journal’s standards. Below, I summarize the reviewers’ comments and provide specific guidance for the necessary revisions.

**Essential Revisions**

**Reviewer 1: Major revisions**

            1.         **Thoracic Temperature Comparisons** :

            •          Reviewer 1 suggests excluding the comparison between thoracic and environmental temperatures due to the lack of statistical differences and the expectation of similarity given dragonflies’ thermoconforming behavior.

            •          **Action** : Consider removing this analysis or providing stronger justification for its inclusion. Reframe the discussion to align with the observed patterns and known thermophysiological behaviors of Odonata.

            2.         **Data Transparency** :

            •          The reviewer highlights the absence of raw environmental temperature data used in the analysis.

            •          **Action** : Ensure all datasets used in the study are shared as supplementary material or through a public repository to align with open science principles.

            3.         **Experimental Design Clarity** :

            •          Provide more detail about the microhabitat sampling and ensure the methodological framework is robust and transparent.

            •          **Action** : Clarify how data from different sites and conditions were standardized for the analyses and address the reviewer’s concerns regarding model assumptions.

**Reviewer 2: Minor Revisions**

The comments focus on grammatical consistency, clarity of explanations, and map visualization. Below are the critical points to address:

            1.         **Grammatical and Structural Edits** :

            •          Multiple minor corrections are noted (e.g., line spacing, phrasing, and syntax).

            •          **Action** : Implement the specific corrections listed, ensuring consistency and readability throughout the manuscript.

            2.         **Statistical Validation** :

            •          Reviewer 2 notes the importance of overdispersion and collinearity diagnostics for the GLMMs.

            •          **Action** : Include a detailed description of the diagnostic procedures and validate the assumptions of your statistical models.

            3.         **Map Revisions** :

            •          The current map outlines only Pará, despite the focus on the Amazon region.

            •          **Action** : Update the map to highlight the broader Amazon region and ensure sampling sites are accurately represented within the hydrographic network.

            4.         **Biological Interpretation** :

            •          Provide additional interpretation of SCBD values and clarify their biological significance concerning thermal strategies and habitat preferences.

**Optional Revisions**

            •          Enhance the introduction by concisely explaining niche breadth and position hypotheses.

            •          Consider simplifying sections with redundant information to improve the manuscript’s flow.

Thank you for your efforts in advancing our understanding of Amazonian Odonata ecology. We look forward to receiving your revised manuscript.

Kind regards,

Gleison Robson Desidério

Academic Editor

PLOS ONE

Journal Requirements: When submitting your revision, we need you to address these additional requirements. 1. Please ensure that your manuscript meets PLOS ONE's style requirements, including those for file naming. The PLOS ONE style templates can be found at https://journals.plos.org/plosone/s/file?id=wjVg/PLOSOne_formatting_sample_main_body.pdf and https://journals.plos.org/plosone/s/file?id=ba62/PLOSOne_formatting_sample_title_authors_affiliations.pdf 2. Please include your full ethics statement in the ‘Methods’ section of your manuscript file. In your statement, please include the full name of the IRB or ethics committee who approved or waived your study, as well as whether or not you obtained informed written or verbal consent. If consent was waived for your study, please include this information in your statement as well. 3. Please include captions for your Supporting Information files at the end of your manuscript, and update any in-text citations to match accordingly. Please see our Supporting Information guidelines for more information: http://journals.plos.org/plosone/s/supporting-information.

**Additional Editor Comments:**

Your manuscript investigates an important and understudied area, but both reviewers have identified key aspects that need attention. The focus on niche breadth, thoracic temperature, and abundance relationships is compelling, yet some methodological choices and interpretations require clarification or refinement.

Reviewer's Responses to Questions

**Comments to the Author**

1. Is the manuscript technically sound, and do the data support the conclusions?

Reviewer #1: Partly

Reviewer #2: Yes

2. Has the statistical analysis been performed appropriately and rigorously? 

Reviewer #1: Yes

Reviewer #2: Yes

3. Have the authors made all data underlying the findings in their manuscript fully available?

Reviewer #1: No

Reviewer #2: Yes

4. Is the manuscript presented in an intelligible fashion and written in standard English?

Reviewer #1: Yes

Reviewer #2: No

5. Review Comments to the Author

Reviewer #1: Dear authors,

The manuscript is a great effort to link thermal performance, functional traits, and beta biodiversity.

I don´t understand why you expected differences between environmental and body temperatures on dragonflies, they are known to be thermoconformers (even with the strategies you describe to control temperature). It is evident from the general comparison you presented that there are no statistical differences between environmental and dragonfly temperatures (Thorax temperature 33.9 SD = 3.7, air temperature 31.6 SD=2.0 for Anisoptera, 29.5 SD 1.6, air temperature 30.1 SD=1.3 ).

It is known that Anisoptera can tolerate higher temperatures due to their behavior and physiological mechanisms to keep their bodies cold. So, the results you found when mixing all the samples in a single model show this. Still, the comparison between the microhabitat and the body temperatures doesn´t make sense since they are expected to be very close.

Finally, you are not sharing the raw data for the environmental temperature used in the analysis.

My recommendation is to keep the niche breadth analysis, exclude the comparison between thoracic temperature and environmental temperature, and rewrite the manuscript.

Reviewer #2: The manuscript presents a relevant investigation into the relationship between morphometry, niche characteristics, and ecophysiology of Odonata in Amazonian streams, providing valuable insights for the ecology and conservation of the group. The approach is methodologically robust, with clear hypotheses supported by detailed field data, appropriate statistical analyses, and literature-based discussions. However, the text requires improvements in grammatical consistency and clarity in certain sections. Given the manuscript's publication potential, I provide minor suggestions that can contribute to a better understanding of the study's results and implications.

The first paragraph introduces an important idea but is somewhat fragmented. Consider explaining the niche breadth and position hypotheses more concisely. For example: “The niche breadth hypothesis suggests that generalist species, with greater tolerance to environmental variations, tend to have higher abundance, whereas the niche position hypothesis argues that species in broadly available habitats are also more abundant.”

Line 44: Citations should be enclosed in parentheses. Also, verify the numerical sequence of citations [1-4]...(11)?

Line 95: Double space: with an annual.

Line 100: Are sample sites spatially independent? “The 46 sampled streams ranged from 1st to 3rd order.”

Lines 120-123: Clarify whether body temperature measurements were taken in a shaded area or under sunlight.

Line 132: Remove the comma: within 10 m of channel,; retention.

Line 134: I understand the importance of the index, but to streamline the text, I suggest removing this sentence: “Each item presented four to six alternatives corresponding to the observed condition related to habitat integrity.”

Line 156: The use of SCBD is well-justified and connected to the objectives, but more detailed descriptions of how the values are biologically interpreted are needed. For example, how do high SCBD values correspond to different thermal strategies or habitat preferences?

Line 167: Add a comma: “For this model we used the Log linkage function.”

Line 168: The analysis mentions the visual validation of GLMMs using the simulated envelope. However, it is crucial to ensure that other fit diagnostics, such as overdispersion and variable collinearity, were also assessed.

Line 177: “How many families? Which are the most abundant species per suborder?”

Line 195: Double space: “above and 4º C”.

Line 222: “Except for niche in Anisoptera” should be “Emerges as an”.

Line 233: “Emerged as ‘a’ predictor.”

Line 252: Error: “N. luzmarinas uch as E. fusca…”.

Line 261: Confusing phrasing: “with abundance e and com Odonata SCBD”. Consider revising for clarity.

Line 270: “Wood?” (Should this be “Bank angle, wood in the stream bed”?)

Lines 270-271: “physicals and chemical” should be corrected to “physical and chemical.” Also, should “Dossel” be “Canopy”?

Line 280: The phrase “make this species with higher thoracic temperatures are more difficult to collect” is confusing. Rewriting for clarity is recommended.

Line 285: “bellow” should be “below.” The phrase “They can be conformers or in some cases and the largest being heliothermic” is unclear and could be rephrased for better understanding.

Line 287: Double space: “maintained 3ºC above and 4 ºC below.”

Line 296: “ecophysiological traits also was” should be “ecophysiological traits also were.”

Line 302: Replace “thermorregulation” with “thermoregulation.”

In the title of the article, you highlight the Amazon, but in the main research map (Fig. 2), you outline only the state of Pará. I recommend outlining the Amazon region. Many sampling sites are located outside the hydrographic net.

6. PLOS authors have the option to publish the peer review history of their article (what does this mean? ). If published, this will include your full peer review and any attached files.

**Do you want your identity to be public for this peer review?** For information about this choice, including consent withdrawal, please see our Privacy Policy .

Reviewer #1: No

Reviewer #2: No

---

## [Author Response · Author response to Decision Letter 1]

30 Mar 2025

UNIVERSIDADE FEDERAL DO PARÁ

Programa de Pós-graduação em Ecologia

Belém, March 24th, 2025

Dear Gleison Robson Desidério,

We are submitting the revised version of the manuscript “Thorax temperature and niche specificity as predictors of Odonata (Insecta) abundance and diversity in Eastern Amazon streams”, (number: PONE-D-23-02245) to PLoS One.

We addressed all the comments made by the reviewers and justified the few cases where the requests could not be attended (please see our specific answers below). We would like to thank the anonymous reviewers for their contributions, which were all very relevant. In addition, we appreciate the swiftness and seriousness of your evaluation process. All changes made to the text are highlighted in red to facilitate the review process.

The comments substantially improved our manuscript, and we hope the revised manuscript is suitable for publication in PLoS One. However, we are willing to keep working if you or the reviewers think that further amendments are needed.

Sincerely,

Lenize Calvão, on the behalf of all authors

Essential Revisions

Reviewer 1: Major revisions

1. Thoracic Temperature Comparisons:

• Reviewer 1 suggests excluding the comparison between thoracic and environmental temperatures due to the lack of statistical differences and the expectation of similarity given dragonflies’ thermoconforming behavior.

• Action: Consider removing this analysis or providing stronger justification for its inclusion. Reframe the discussion to align with the observed patterns and known thermophysiological behaviors of Odonata.

Author’s response: We appreciate the reviewer’s comment and understand the concern regarding the statistical relevance and biological interpretation of the thorax-air temperature comparison. However, we believe that retaining this analysis is important, as it provides valuable ecophysiological insights into the thermoregulation strategies adopted by different suborders.

Specifically, Figure 4 illustrates clear deviations between thoracic and ambient temperatures. For example, Erythemis castanea (Anisoptera) reached up to 8°C above air temperature, while most Anisoptera species maintained a difference of approximately 2°C above air temperature. Note that also larger Zygoptera showed deviations of up to 4°C below air temperature, while the majority remained 1–2°C below. These patterns reflect the thermoregulatory continuum described in the literature (e.g., May 1976; Corbet & May 2008), and our data support the distinction between thermal conformers and heliothermic/endothermic species. Although the difference alone may not be statistically significant in isolation, it complements the significant effects found in Table 2 and strengthens the ecological interpretation of thorax width and air temperature as predictors of body temperature.

Thus, we respectfully request to keep this analysis, as it contributes to identifying thermal guilds and interpreting the functional ecology of Odonata in Amazonian streams.

2. Data Transparency:

• The reviewer highlights the absence of raw environmental temperature data used in the analysis.

• Action: Ensure all datasets used in the study are shared as supplementary material or through a public repository to align with open science principles.

Author’s response: Thank you for this important observation. In response, we have included the raw environmental temperature data (along with other relevant variables) as Supplementary Material (S4 Table) in the revised submission. This ensures full transparency and reproducibility of our analyses, in accordance with PLOS ONE's data sharing policies and open science principles.

3. Experimental Design Clarity:

• Provide more detail about the microhabitat sampling and ensure the methodological framework is robust and transparent.

• Action: Clarify how data from different sites and conditions were standardized for the analyses and address the reviewer’s concerns regarding model assumptions.

Author’s response: Thank you for your comment. In the revised version of the manuscript, we have clarified how environmental and biological data were standardized prior to statistical analyses. Specifically, we added the following sentence (Lines 173–174): Line 173-174: To assess the effect of thoracic temperature, niche breadth and position, predictor variables were standardized using decostand function (method = 'standardize') from the vegan package in R.

Reviewer 2: Minor Revisions

The comments focus on grammatical consistency, clarity of explanations, and map visualization. Below are the critical points to address:

1. Grammatical and Structural Edits:

• Multiple minor corrections are noted (e.g., line spacing, phrasing, and syntax).

• Action: Implement the specific corrections listed, ensuring consistency and readability throughout the manuscript.

Author’s response: Thank you very much for your careful review and for pointing out grammatical inconsistencies and minor structural issues. We have thoroughly revised the manuscript to correct all typographical, grammatical, and formatting issues, including line spacing and syntax. All changes are marked in red text in the revised version to facilitate your review.

2. Statistical Validation:

• Reviewer 2 notes the importance of overdispersion and collinearity diagnostics for the GLMMs.

• Action: Include a detailed description of the diagnostic procedures and validate the assumptions of your statistical models.

Author’s response: Thank you for highlighting the importance of validating model assumptions. We addressed your concern by conducting both collinearity and overdispersion diagnostics for all models, and we have included these details in the revised manuscript.

We tested the correlation between all predictor variables (Thorax temperature, Niche position, and Niche breadth). None of the pairwise correlations exceeded 0.50, indicating no problematic collinearity:

Thorax temperature (ºC) Niche position (NP) Niche breadth (NB)

Niche position (NP) 0.27

Niche breadth (NB) -0.12 -0.46

Abundance -0.13 -0.48 0.79

Abundance was initially considered as a predictor for SCBD, but due to its high correlation with Niche breadth (0.79), it was excluded from the final models to avoid multicollinearity.

We assessed overdispersion following the recommendations of Zuur et al. (2009). Models in Tables 2 and 3 showed no evidence of overdispersion, with dispersion indices of: Dispersion index 0.957, 0.983 (Zygoptera table 3) and 0.945 (Anisoptera table 3). Above 1 the model is considered overdispersed according to Zuur. (Zuur AF, Ieno EN, Walker NJ, Saveliev AA, Smith GM. Mixed effects models and extensions in ecology with R. New York: Springer Science & Business Media; 2009).

In Table 4, some overdispersion was detected despite using a negative binomial distribution: for Odonata (1.058), Anisoptera (1.200) and Zygoptera (1.217).

To ensure model validity, we evaluated the goodness-of-fit using simulated envelope plots from the hnp package (Moral et al., 2017). These plots confirmed that residuals fell within the 95% confidence intervals, supporting the reliability of the fitted models.

Please note that due to persistent overdispersion in the full Odonata dataset, and lack of fit in the GLMM validation, we removed this model from the current version and retained only those with suborder-specific analyses (Anisoptera and Zygoptera), which had acceptable diagnostic performance through hnp evaluation.

We hope these clarifications adequately address the reviewer’s concern and reinforce the robustness of our statistical approach.

Anisoptera Zygoptera

3. Map Revisions:

• The current map outlines only Pará, despite the focus on the Amazon region.

• Action: Thank you for this valuable suggestion. In the revised version of the manuscript, we updated Figure 2 to improve the spatial representation of our study area. We hope this improved visualization enhances the reader’s understanding of the study's spatial scope and environmental context.

4. Biological Interpretation:

• Provide additional interpretation of SCBD values and clarify their biological significance concerning thermal strategies and habitat preferences.

Author’s response: Thank you for this insightful suggestion. In the revised version of the manuscript, we improved the interpretation of Species Contribution to Beta Diversity (SCBD) by explicitly linking these values to species’ thermoregulatory strategies and habitat preferences. These improvements are reflected in both the Hypotheses section and the Discussion.

We clarified that species with high SCBD tend to occupy specialized microhabitats or exhibit restricted thermal tolerances, which may cause them to occur only under specific environmental conditions. Conversely, generalist species with broader thermal tolerance and greater dispersal capacity can occur across a wider environmental gradient, which also results in a strong influence on compositional variation between sites.

In the Discussion, we expanded on how thermal guilds, inferred from thoracic temperature data, relate to patterns of SCBD — highlighting, for example, that heliothermic Anisoptera and conformer Zygoptera contribute differently to spatial turnover depending on their niche breadth and habitat specificity.

These additions aim to strengthen the ecological interpretation of SCBD and its relevance in understanding the functional composition of Odonata assemblages in Amazonian streams.

Optional Revisions

• Enhance the introduction by concisely explaining niche breadth and position hypotheses.

• Consider simplifying sections with redundant information to improve the manuscript’s flow.

Author’s response: Thank you for these thoughtful suggestions. In the current version, we have ensured that the niche breadth and niche position hypotheses are clearly and concisely explained in the Introduction (Lines 10–16), following classical ecological theory. We also revised the manuscript to eliminate redundant phrasing and improve readability throughout. At this stage, we believe the manuscript is well-structured and flows appropriately, and no further changes were necessary. We appreciate your guidance in helping us refine the text.

Reviewer #1: Dear authors,

The manuscript is a great effort to link thermal performance, functional traits, and beta biodiversity.

I don´t understand why you expected differences between environmental and body temperatures on dragonflies, they are known to be thermoconformers (even with the strategies you describe to control temperature). It is evident from the general comparison you presented that there are no statistical differences between environmental and dragonfly temperatures (Thorax temperature 33.9 SD = 3.7, air temperature 31.6 SD=2.0 for Anisoptera, 29.5 SD 1.6, air temperature 30.1 SD=1.3 ).

It is known that Anisoptera can tolerate higher temperatures due to their behavior and physiological mechanisms to keep their bodies cold. So, the results you found when mixing all the samples in a single model show this. Still, the comparison between the microhabitat and the body temperatures doesn´t make sense since they are expected to be very close.

My recommendation is to keep the niche breadth analysis, exclude the comparison between thoracic temperature and environmental temperature, and rewrite the manuscript.

Author’s response: We appreciate the reviewer’s comment and understand the concern regarding the statistical relevance and biological interpretation of the thorax-air temperature comparison. However, we believe that retaining this analysis is important, as it provides valuable ecophysiological insights into the thermoregulation strategies adopted by different suborders.

Specifically, Figure 4 illustrates clear deviations between thoracic and ambient temperatures. For example, Erythemis castanea (Anisoptera) reached up to 8°C above air temperature, while most Anisoptera species maintained a difference of approximately 2°C above air temperature. Note that also larger Zygoptera showed deviations of up to 4°C below air temperature, while the majority remained 1–2°C below. These patterns reflect the thermoregulatory continuum described in the literature (e.g., May 1976; Corbet & May 2008), and our data support the distinction between thermal conformers and heliothermic/endothermic species. Although the difference alone may not be statistically significant in isolation, it complements the significant effects found in Table 2 and strengthens the ecological interpretation of thorax width and air temperature as predictors of body temperature.

Thus, we respectfully request to keep this analysis, as it contributes to identifying thermal guilds and interpreting the functional ecology of Odonata in Amazonian streams.

Finally, you are not sharing the raw data for the environmental temperature used in the analysis.

Author’s response: Thank you for your thoughtful and constructive feedback. We appreciate your recognition of our effort to integrate thermal performance, functional traits, and beta diversity.

We would like to clarify that the values referenced in Table 1 are descriptive statistics (mean ± SD), presented only as a general summary of thoracic and air temperatures across suborders. These values are not the basis of any statistical test, and we have updated them in the revised version to reflect corrected averages.

The statistical comparisons relevant to this point are provided in Tables 2 and 3, where we used generalized linear mixed models (GLMMs) to test the relationships between thoracic temperature, air temperature, and thorax width. These models showed significant differences and interactions, particularly when considering each suborder independently. For example:

• Anisoptera often exhibited thoracic temperatures above air temperature, with individual species (e.g., Erythemis castanea) reaching up to 8°C higher, while most were around 2°C above;

• Zygoptera typically maintained thoracic temperatures 1–2°C below air temperature.

These differences, although small in some cases, are biologically meaningful and support the existence of a continuum of thermoregulatory strategies rather than a strict classification of all Odonata as thermoconformers. Our goal is to identify these species-level differences that may relate to behavior, body size, and habitat use.

Therefore, we believe that retaining the comparison between thoracic and environmental temperatures, as implemented through model-based tests, is important to illustrate variation in thermal strategies, which complements the niche breadth analysis.

Finally, in response to your concern about data transparency, we have included all raw environmental temperature data as Supplementary Material (S4 Table) in this revised submission.

We hope this clarifies our rationale and reinforces the contribution of this comparison to understanding the ecophysiological variation among Amazonian Odonata.

Reviewer #2:

The manuscript presents a relevant investigation into the relationship between morphometry, niche characteristics, and ecophysiology of Odonata in Amazonian streams, providing valuable insights for the ecology and conservation of the group. The approach is methodologically robust, with clear hypotheses supported by detailed field data, appropriate statistical analyses, and literature-based discussions. However, the text requires improvements in grammatical consistency and clarity in certain sections. Given the manuscript's publication potential, I provide minor suggestions that can contribute to a better understanding of the study's results and implications.

The first paragraph introduces an important idea

---

## [Editor Report · Decision Letter 1]

Thorax temperature and niche characteristics as predictors of abundance of Amazonian Odonata

PONE-D-24-40484R1

Dear Authors,

We’re pleased to inform you that your manuscript has been judged scientifically suitable for publication and will be formally accepted for publication once it meets all outstanding technical requirements.

Kind regards,

Gleison Robson Desidério

Academic Editor

PLOS ONE

Additional Editor Comments (optional):

Dear Authors,

I am pleased with the revised version of the manuscript and appreciate the authors’ careful attention to the corrections suggested in the first round of review. These revisions have significantly improved the quality of the work, which was evident during my evaluation.

I am confident that the manuscript is now suitable for publication in PLOS One.

Sincerely,

Gleison Desidério
---

## [Editor Report · Acceptance letter]

PONE-D-24-40484R1

PLOS ONE

Dear Dr. Córdoba-Aguilar,

I'm pleased to inform you that your manuscript has been deemed suitable for publication in PLOS ONE. Congratulations! Your manuscript is now being handed over to our production team.

Kind regards,

on behalf of

Dr. Gleison Robson Desidério

Academic Editor

PLOS ONE